# Effect of Adding Probabilistic Zonal Prior in Deep Learning-based Prostate Cancer Detection

**Matin Hosseinzadeh**                    Matin.Hosseinzadeh@radboudumc.nl
**Patrick Brand**                    Patrick.Brand@radboudumc.nl
**Henkjan Huisman**                    Henkjan.Huisman@radboudumc.nl
*Diagnostic Image Analysis Group, Radboudumc, Nijmegen, The Netherlands*

## Abstract

We propose and evaluate a novel method for automatically detecting clinically significant prostate cancer (csPCa) in bi-parametric magnetic resonance imaging (bpMRI). Prostate zones play an important role in the assessment of prostate cancer on MRI. We hypothesize that the inclusion of zonal information can improve the performance of a deep learning based csPCa lesion detection model. However, segmentation of prostate zones is challenging and therefore deterministic models are inaccurate. Hence, we investigated probabilistic zonal segmentation. Our baseline detection model is a 2DUNet trained to produce a csPCa heatmap followed by a 3D detector. We experimented with the integration of zonal prior information by fusing the output of an anisotropic 3DUNet trained to produce either a deterministic or probabilistic map for each prostate zone. We also investigate the effect of early or late fusion on csPCa detection. All methods were trained and tested on 848 bpMRI. The results show that fusing zonal prior knowledge improves the baseline detection model with a preference for probabilistic over deterministic zonal segmentation.

**Keywords:** Prostate cancer, Detection, Deep learning

## 1. Introduction

Prostate cancer is the second most frequent cancer and the fifth leading cause of cancer death in men (Bray et al., 2018). Early detection of prostate cancer can decrease the mortality rate and make the disease treatable. In recent years, prostate MRI has demonstrated the ability of prostate cancer diagnosis and now it is one of the main imaging tools for detecting prostate cancer in clinic(van der Leest et al., 2019). However, diagnosing and grading of prostate cancer lesions using MR images is difficult and requires substantial expertise(Rosenkrantz et al., 2016).

Computer-Aided Detection (CAD) systems can help radiologists by automatically detecting the clinically significant prostate cancer (csPCa) lesions in MR images, but current CAD systems are still performing below the expert level. Prostate CAD systems may be improved by including zonal segmentations, since the prostate zones play a crucial role in diagnostic process in clinic because the occurrence and appearance of prostate cancer are dependent on its zonal location. The two main areas of interest in the prostate are the transition zone (TZ) and the peripheral zone (PZ). The PZ is the area where most clinically significant prostate cancer lesions grow, approximately 70%-75% of prostate cancers originate in the PZ and 20%-30% in the TZ (Weinreb et al., 2016). Moreover, cancers of these two zones exhibit different behaviors. Based on the location of the lesion, different

MRI modalities are majorly used for determining the type of the lesion. For the PZ, DWI is the primary determining sequence, but for the TZ, T2W is primary determining sequence (Weinreb et al., 2016). However, automatic prostate zonal segmentation is challenging since the boundaries especially at the base, apex and the interface of the zones are usually ambiguous and consequently, the accuracy of automatic segmentations methods especially at PZ is very low (Dice similarity coefficient 0.67) (de Gelder and Huisman, 2018).

We hypothesize, by incorporating computer-generated prostate zonal probabilistic segmentation as prior knowledge to a deep learning model we can improve csPCa lesion detection performance.

## 2. Methods

### 2.1. Data

Data used in this study was a local, retrospective dataset of consecutive bpMRI (T2W and computed ADC and high b-value DWI) of 848 patients. In this data, lesions were clinically reported by expert radiologists using PIRADSv2 (Weinreb et al., 2016). As a model of csPCa, all PIRADS 4 and 5 lesions were selected and manually delineated by 2 students. In total 319 patients have at least one csPCa lesion. All images were re-sampled to 0.5x0.5x3.6mm resolution and cropped by 9.6x9.6cm around the center. Training, validation and test sets were generated by randomly distributing the patients in a ratio of 3:1:1 through stratified sampling. Thus, they had non-overlapping patients and equal distribution of patients with csPCa was in each set.

### 2.2. Experiments

All networks were trained for 200 epochs using Adam optimizer and weighted cross entropy loss, with a learning rate of $10^{-5}$. Data augmentation was applied during the training phase to reduce overfitting. All predicted 2D heatmaps of a patient were combined to create a 3D volumetric heatmap, on which a two-threshold method was applied to segment csPCa lesions with a probability score for each of them. For model selection, we selected the best validation model based on the average sensitivity at several points on Free-response ROC (FROC) curves. Figure 1 gives an overview of the proposed method.

**Experiment 1 - Baseline:** We trained a 2DUnet (Ronneberger et al., 2015) with 3 input channels to segment/detect the lesions.

**Experiment 2 - Early Fusion:** As the manual prostate zonal segmentations were not available for our dataset we used a modified 3DUnet for anisotropic images (Mooij et al., 2018), which were trained using 53 T2W images and manual zonal segmentations, to generate prostate probabilistic and deterministic zonal segmentations for all cases in this study. We used these zonal segmentations as extra channels at the input of the baseline model.

**Experiment 3 - Late Fusion:** We did an experiment same as Experiment 2 but instead of using zonal segmentations as inputs of the model, we combined them to the last feature map of the UNet before the 1x1 convolution layer.

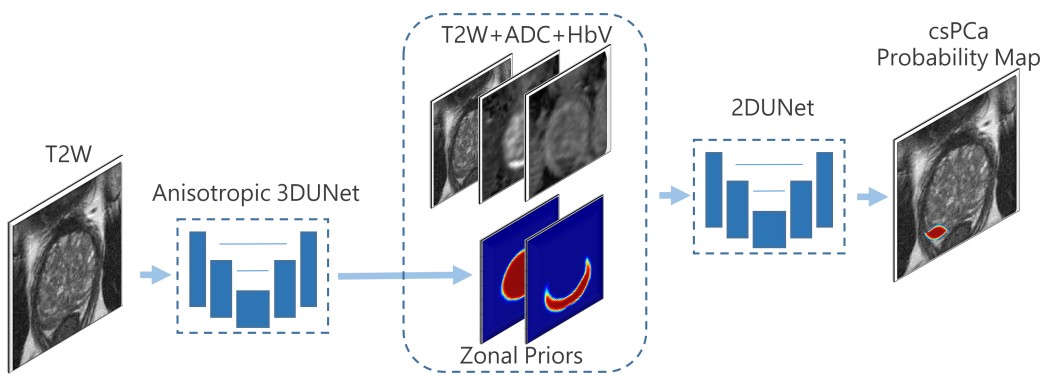

Figure 1: Schematic overview of the method

Table 1: Sensitivity at various FPs per patient on the test set

| Model | Sens@0.5FP | Sens@1FP | Sens@2FPs | Average |
|---|---|---|---|---|
| Baseline | 0.760 | 0.825 | 0.825 | 0.803 |
| Early Fusion - Probabilistic | **0.795** | **0.887** | **0.887** | **0.856** |
| Late Fusion - Probabilistic | 0.774 | 0.873 | 0.873 | 0.840 |
| Early Fusion - Deterministic | 0.774 | 0.802 | 0.802 | 0.793 |
| Late Fusion - Deterministic | 0.774 | 0.816 | 0.816 | 0.802 |

## 3. Results and Discussions

Table 1 shows that the models trained using probabilistic zonal segmentations achieved better performance on the lesion detection task. Particularly these models find more csPCa lesions in the same FP rate compared to the baseline and deterministic models.

Well trained deep learning networks with sufficient data are meant to automatically learn most useful features in the data such as prostate zones information. However, in the medical imaging domain the data and annotations are scarce and providing abundant data is usually impossible. As a result, providing the deep learning model with prior knowledge can be beneficial. In this paper, we showed that, given the same size of training data, providing prior knowledge using a two-stage approach can help a detection model to perform better compared to a single model which could not capture all prostate zonal information automatically. Moreover, we showed that when deterministic segmentation is challenging, probabilistic segmentations can be more beneficial for providing prior knowledge.

In conclusion the results demonstrate that the proposed early fusion of probabilistic segmentations method achieves the best results among the compared methods in this paper.

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
