# OpenReview forum: "Effect of Adding Probabilistic Zonal Prior in Deep Learning-based Prostate Cancer Detection"
_MIDL.io/2019/Conference/Abstract — MIDL Abstract 2019_

### Official Review · AnonReviewer2 · 2019-04-30

**Rating:** 3
**Confidence:** 3

**Review:**

This work proposes to integrate prior knowledge into a segmentation network to combat data scarcity. The experiments are conducted on a large dataset, with promising and sound performance demonstrated. The writing is clear and easy to follow.

---

### Official Review · AnonReviewer1 · 2019-05-01
**Useful approach and strong evaluation. limited novelty.**

**Rating:** 3
**Confidence:** 2

**Review:**

The abstract explores different ways of using segmentations as a spatial prior in prostate lesion detection: early and late fusion; probabilistic and deterministic segmentations.
Results with probabilistic segmentations show convincing improvements.  This is a practical way of incorporating such information that can be useful in many applications and this work nicely illustrates that.
The main strength of the work is the validation on a large dataset of bi-parametric MRI (848 patients) with clinically reported lesions by expert radiologists using PIRADSv2.

How was cross-entropy weighted? I wondered if the poor performance of incorporating the deterministic segmentations could be related also to undersegmentation of the structures due to class imbalance.

---

### Decision · Program_Chairs · 2019-05-06
**Acceptance Decision**

Accept